# Mesoscale Analysis of the Effect of Interfacial Transition Zone on the Compressive Damage of Concrete Based on Discrete Element Method

**DOI:** 10.3390/ma15248840

**Published:** 2022-12-10

**Authors:** Jian-Jun Lei, Ze-Xiang Wu, Zheng-Jun Wen, Zi-Shan Cheng, Ran Zhu

**Affiliations:** 1College of Architecture and Civil Engineering, Wenzhou University, Wenzhou 325000, China; 2Wenzhou Public Utilities Development Group Co., Ltd., Wenzhou 325000, China; 3Shanghai Construction Group Co., Ltd., Shanghai 200080, China

**Keywords:** axial compression test, interface transition zone, crack development, particle contact, discrete element method

## Abstract

The coarse aggregate–mortar interface transition zone (ITZ) has a great influence on the mechanical properties of concrete, which cannot be easily studied using laboratory tests in the mesoscale. In this paper, a series of axial compression tests were conducted using the discrete element method (DEM) on concrete specimens for four phases: coarse aggregates, mortars, aggregate–mortar interface transition zones, and voids. The effects of ITZ strength on macroscopic stress and microscopic cracks under different strength reduction factors were investigated through axial compression testing. With the increase in interface transition strength, the compressive strength of the concrete becomes stronger; moreover, the number of cracks decreases, and the anisotropy of contact orientation becomes weaker. Meanwhile, the direction of crack development and the damage mode of compressed concrete specimens were also dependent on the coarse aggregate–mortar interface strength coefficient.

## 1. Introduction

As the material with the highest strength-to-price ratio, concrete has played a revolutionary role in the world’s architectural history over the past century. In 1965, Fanran [1] observed the interface transition zone (ITZ) between the mortar and coarse aggregate on concrete with optical microscope for the first time. The thickness of this special zone normally ranges from 40 to 50 μm [2] and fills with increased residual porosity after the cement exothermic reaction [3]. The mesoscopic features of these complex structures associated with strength degradation in the zone between coarse aggregates and mortars play a significant role in the study of concrete’s mechanical properties.

The contributing factors to the properties of the interface transition zone have been studied widely. For example, silica fume can fill the holes created by the cement’s exothermic reaction and improve the density of the interface between the aggregate and mortar [4,5]. An admixture of metakaolin and limestone powder can reduce the thickness of the interface transition zone, and fly ash can slow down the hardening of the ITZ [6]. Instead, if the volume of the aggregates is 75–80% and the concrete is compacted without stirring, the thickness of the interface transition zone will be increased [7,8]. Moreover, pre-treatment of the coarse aggregate will also affect the ITZ’s properties [9]. Due to the above-mentioned influencing parameters, the strength of ITZ varies in each specimen. Even if in the same specimen, the ITZ in different areas also have different levels of thickness and hardness; thus, it is a challenge to keep the ITZ strength strictly consistent in laboratory preparation, let alone conduct tests on it. Therefore, this paper uses the numerical simulation method to establish a model that considers the coefficient of aggregate–mortar interface strength to avoid problems with preparation.

Among the many simulation methods, the discrete element method (DEM) proposed by Cundall in 1971 [10,11] has unique advantages in simulating concrete failure. Sun et al. [12] suggested that the size effects of concrete specimens can be avoided by keeping the ratio of the length of the specimen’s side over the minimum particle radius of at least 80; moreover, the authors also examined the size effect of the concrete specimens and rate effect of the loading speed on the performance of the stress–strain curve based on DEM [13,14]. During the numerical simulation, the outlines of the concrete coarse aggregate were also collected via digital camera [15] and CT scanning technology [16,17] to better reproduce the boundary conditions of the ITZ between aggregates and mortars. Moreover, the mesoscale analyses emphasised the initial defects in the concrete fractures [18,19] and mentioned the role of ITZ in the evolutions of cracks in concrete specimens and the corresponding principle of damage.

At present, the main analysis points of ITZ effect are focused on macro mechanical properties [15,20,21], such as stress–strain curve, destruction, etc., and there is a lack of analysis on the internal contact force of granular materials. In this study, the four phases of concrete specimen embedding—coarse aggregates, mortars, ITZ, and voids—were first established by DEM based on the commercial software PFC2D. The stress–strain performance of the concrete model was then validated via laboratory tests. The following parametric studies were also conducted to investigate the mechanical characteristics of macroscopic stress and microscopic cracks considering the interface strength between aggregate and mortar.

## 2. DEM Modelling of Concrete Specimen

### 2.1. Contact Model

The parallel bond model (PBM) developed by D.O. Potion and P.A. Cundall [22] was used in this present study, which included bonded and unbonded parts, as shown in Figure 1a. The model theory is explained in Figure 1b, in which the unbonded part incorporated the contact parameters of normal stiffness ($k_{n}$), tangential stiffness ($k_{s}$), contact activation ($g_{s}$), and the friction coefficient (μ), and the latter parts consisted of the bond parameters of the normal stiffness (${\bar{k}}_{n}$), tangential stiffness (${\bar{k}}_{s}$), maximum tensile strength, and (${\bar{\sigma }}_{c}$) maximum shear strength (${\bar{\tau }}_{c}$). Moreover, Figure 1c presents the strength criterion of PBM. When the bond-related tensile stress $\bar{\sigma }$ exceeds the maximum tensile strength ${\bar{\sigma }}_{c}$, the concrete shows tensile failure; when the shear stress $\bar{\tau }$ exceeds the maximum shear strength ${\bar{\tau }}_{c}$, the concrete shows shearing failure. The tensile and shear stresses acting on the bond can be calculated using the following equation:(1)$$\{\begin{array}{c} \bar{\sigma }=\frac{-{\bar{F}}_{n}}{A}+\frac{\left|\bar{M_{s}}\right|R}{I}<{\bar{\sigma }}_{c}\\ \bar{\tau }=\frac{{\bar{F}}_{s}}{A}+\frac{\left|\bar{M_{n}}\right|R}{J}<{\bar{\tau }}_{c} \end{array}$$
where ${\bar{F}}_{n}$, ${\bar{F}}_{s}$ and ${\bar{M}}_{n}$, ${\bar{M}}_{s}$ denote the axial- and shear- directed forces and moments, respectively, and *R*, *A*, *I*, and *J* are the radius, area, moment of inertia, and polar moment of inertia of the bond cross sections, respectively.

### 2.2. Generation of Concrete Specimen

Herein, the axial laboratory compression test on the concrete specimen with a size of 50 × 50 × 50 mm^3^ was selected as a reference experiment [15], in which the concrete was damaged at a 0.12% axial strain corresponding to a peak stress of 29 MPa due to the connection of cracks between the upper and lower boundaries. Accordingly, a two-dimensional DEM model of concrete with four phases—mortars, coarse aggregates, ITZs, and voids [23,24,25] was then established. In this model, a total of 27,216 particles with a radius ranging from 0.5 mm to 2.5 mm were randomly seeded as mortar in a 50 mm × 50 mm space, and the unbalanced force inside the mortar reached a minimal level by the wall servo. The coarse aggregates and voids were then modelled according to the outline of the cutting surface of the concrete specimen. The cutting surface with the outline of coarse aggregates are shown in Figure 2a [15]. The concrete discrete element model can finally be completed after eliminating the unbalanced force by servo boundary conditions, as shown in Figure 2b. Furthermore, three types of PBM bonding properties can be generated based on two types of particles—coarse aggregates and mortars—as shown in Figure 3.

### 2.3. Model Validation

The mechanical parameters of concrete components show a certain relationship (48% of coarse aggregates strength = 80% of mortars strength = ITZ strength), according to parameters calibrated by previous studies [15,17,26]. The single material parameters include effective modulus (*E*), stiffness ratio (*k*), bonding effective modulus ($\bar{E}$), bonding stiffness ratio ($\bar{k}$), maximum tension strength (${\bar{\sigma }}_{c}$), cohesion ($\bar{c}$), and internal friction angle ($\bar{\phi }$), as shown in Table 1, which uses the parameters of PBM for concrete specimens referenced by Shi et al. [27].

Figure 4a shows the results of the simulation in which the mechanical curve of the real test can be well captured using the proposed DEM model. Figure 4b is the nephogram of the axial stress (*σ*_y_) field in the specimen at an axial strain level of 0.0011, at which the stress in the stress–strain curve reaches maximum of 28.7 MPa. The stress distribution in the specimen is inhomogeneous, and the maximum stress zone accumulated around the outlines of the coarse aggregate, shown in red, is 57.3 MPa. Compared with the peak stress in the stress–strain curve, the maximum stress in the specimen can reach two times the peak stress of the specimen; meanwhile, the weak stress parts also formed around the voids in the specimen, shown in blue in the figure, which demonstrate that the existence of coarse aggregates and voids increases the inhomogeneous stress field and stress concentration in the specimen. Figure 4c shows the crack number–strain curve of the model. In the figure, the cracks are divided into three parts (initial cracks, mid-term cracks, and later cracks), according to the sequence of generation. The growth rate of the cracks in the three parts moved from slow to fast and then to slow, and this change of curvature reflects the brittleness of real concrete. Moreover, Figure 4d shows the failure surface of the concrete model. The cracks caused by tensile fractures contributed to the main failure surface (shown as blue lines), and the local shearing cracks in the mortar are distributed in a small range (shown as red lines). The distribution of tensile cracks is mainly transmitted between coarse aggregates and forms the connection between the upper and lower boundaries.

## 3. Effect of ITZ Strength

The simulations based on the proposed mesoscale DEM concrete model were carried out considering the degradation of the ITZ strength coefficients. The main strength coefficients of the concrete model in PFC are maximum tension strength and cohesion strength (${\bar{\sigma }}_{c}$ and $\bar{c}$), which will be reduced in simulations. As shown in Table 2, the strength coefficients were multiplied by reduction factors (defined as *R*), and the reduction factors are listed as 0.2, 0.4, 0.6, 0.8, and 1.0 in five tests, respectively. The maximum shearing strength (${\bar{\tau }}_{c}$) was also reduced according to the Mohr–Coulomb yield criterion, and the other parameters were constant, according to Table 1. The loading speed of the software was kept at 0.1 m/s, and the calculation was terminated until the axial strain reached 0.2%.

## 4. Analysis of Simulation Results

### 4.1. Effect of Mechanical Strength

Figure 5 shows that peak stress rises with an increase in the reduction factor and trends toward steady until the reduction factor equals 0.8. Notably, the development of stress–strain curves trends toward divergence after the axial strain reaches 0.08% due to the different plastic deformation caused by the parameters of the ITZ’s strength coefficient. Figure 6 shows the axial stress distribution in the specimens under different ITZ strength reduction factors and strain levels, in which increasing the ITZ strength will enlarge the area of the high stress area (red zone) obviously.

### 4.2. Effect of Crack Development

The influence of the ITZ strength coefficient on crack development was analysed based on the number of cracks during the process of stress growth, as shown in Figure 7. The increase in the ITZ strength coefficient reduction factor makes the peak stress stronger and produces a smaller number of cracks at a strain of 0.2%, and a concrete specimen with a bigger ITZ strength coefficient is prone to brittle failure. Figure 7 is the crack elevation–strain curve of the test with *R* = 1, and the slope of the curve is 4.092 × 10^−3^; additional statistical analysis of crack elevation was also performed. The proportions of the different crack elevations inside the concrete specimen at a strain of 0.2% are listed in Figure 7c, wherein the slope of the curve reveals the distribution characteristic of the cracks. The corresponding slope becomes steeper with the increase in the reduction factor, which reflects the horizontal cracks gradually transforming into vertical cracks.

The spatial distribution of the initial cracks, mid-term cracks, and later cracks are displayed in Figure 8a. Under the different reduction factors, the cracks’ spatial distribution is obviously different than with the initial cracks; as strength coefficients increase, the development zone of the initial cracks transforms from an ITZ zone into a void zone. Conversely, the mid-term and later cracks’ distribution scattered and then concentrated in the mortar zone; with the increase in ITZ strength coefficients, the distribution of the cracks is more concentrated and converged along a vertical direction.

In addition, shearing and tension cracks within the concrete specimen at a strain of 0.2% are shown in Figure 8b. For the *R* = 0.2 case, abundant shearing cracks, shown as red lines, formed surrounding the coarse aggregates in the later failure state, which corresponds to a weaker shearing resistance within the specimen. However, the specimen with *R* = 1.0 was damaged by the connection of tension cracks, shown as blue lines, from the bottom to top boundaries.

### 4.3. Effect of Contact Development

Figure 9a shows the distribution of contact force-chains considering the levels of ITZ strength coefficient *R* in a state of axial strain *ε* = 0.12%, at which the axial stress is almost near the peak stress where the distribution of the force-chain in each specimen is clear. The distribution density of the force-chains was found to obviously increase with improvement in the strength coefficient *R*. In addition, the probability distribution of contact force P(*f*) is taken as a mesoscopic characteristic index in which *f* is the ratio of contact force over the mean contact force, and it can be calculated using the following Equation (2):(2)$$P(f)=\frac{N{\left[\;\right]}_{k}}{N},\;k=1,2,\cdots ,N$$
where *N* is the total number of intervals; *k* is the number of intervals; and *N*[ ]*_R_* is the amount of contact force in the *k* interval. For example, [ ]*_k_* is the number of contact force, which when divided by mean contact force ranges from *k*−1 to *k*.

Figure 9b divides the contact force into two parts: the weak contact force interval with *f* < 1 and the strong contact force interval with *f* > 1. It can be clearly seen that with the increase in *R*, the probability of weak contact force decreases and the probability of strong contact force increases. Therefore, the ITZ strength will affect the proportion of the strong contact force-chain inside the specimen, so the internal force transmitted by the concrete increases, corresponding to the stronger load capacity of the concrete.

Contact orientation is the probability of a contact force-chain forming in all directions, which can be described by the density function *E*(*θ*) [28], and *E*(*θ*) represents the proportion of the contact whose direction is within *D θ* around the *E*(*θ*) and has the following properties:(3)$$\int_{0}^{2\pi }E(\theta )d\theta =1$$
(4)E(θ)=E(θ+π)

Due to the periodicity of *E*(*θ*), it can be approximated by a Fourier series:(5)$$E(\theta )=\frac{1}{2\pi }\left[1+a\cdot \cos2\left(\theta -\theta_{a}\right)\right]$$
where *θ_a_* represents the principal direction of normal contact and *a* is the anisotropy coefficient. When *a* = 0, the specimen is anisotropic, and this characteristic will be more obvious as *a* increases.

Hence, the index *a* can examine the level of the anisotropy of each specimen. Integrating the Fourier series of *E*(*θ*) gives:(6)$$\int_{0}^{2\pi }E(\theta )\cos2\theta d\theta =\frac{a}{2}\cos2\theta_{a}$$
(7)$$\int_{0}^{2\pi }E(\theta )\sin2\theta d\theta =\frac{a}{2}\sin2\theta_{a}$$

The anisotropy index *a* can be expressed as:(8)$$a=2\sqrt{{\left(\int_{0}^{2\pi }E(\theta )\cos2\theta d\theta \right)}^{2}+{\left(\int_{0}^{2\pi }E(\theta )\sin2\theta d\theta \right)}^{2}}$$
and *θ_a_* can be expressed as:(9)$$\theta_{a}=0.5\cdot \arctan\frac{\int_{0}^{2\pi }E\left(\theta \right)\sin2\theta d\theta }{\int_{0}^{2\pi }E\left(\theta \right)\cos2\theta d\theta }$$

According to Equations (8) and (9), the anisotropy of each specimen can be obtained. Figure 10 is a summary of the contact anisotropy of the specimen under different reduction coefficients *R* and axial strain *ε*. Overall, the contact orientation of the specimen remains vertical during loading. However, when the strain is 0.2%, the contact force in the specimen is so small that the contact orientation changes significantly.

The anisotropy indices are reflected in Figure 11, and Figure 11a shows the relationship between the contact orientation and axial strain; when the strain is less than or equal to 0.16%, the contact orientation of the specimen is basically kept in the vertical direction (90°), but after the strain exceeds 0.16%, each contact orientation of the specimen has a certain fluctuation. Figure 11b shows the relationship between anisotropy index *a* and strain *ε*. The figure shows that as the strain increases, *a* also increases, especially after plastic strain, since the occurrence of failure will enhance the anisotropic properties of the specimen. Moreover, improving *R* will decrease the anisotropy index *a* of the specimen, indicating that enhancing the ITZ strength will make the particles inside the specimen display more balanced stress in all directions.

## 5. Conclusions

In this paper, a four-phase concrete model based on the real damage surface of concrete was established by PFC2D. The model can reflect the mechanical properties of concrete well by considering the effects of ITZ strength. On this basis, compression tests were carried out with different ITZ strength coefficients.

In the tests, macro mechanical strength curves are first studied; the peak stress of concrete increases as the ITZ strength increases, and the corresponding effects can be ignored after *R* ≥ 0.8, which is the same as previous studies [17,20]. Further study of stress nephogram shows that the stress in the specimen is inhomogeneous and accumulates around outlines of coarse aggregate, and maximum stress in the specimen is around two times the peak stress of the specimen.

The micro crack and contact characteristics are subsequently studied; with the increase in ITZ strength, the initial cracks are transformed from an ITZ zone into a void zone, and the crack’s number and elevation increase. Meanwhile, the concrete is prone to brittle failure and the concrete will be damaged from shearing to tensile. For microscopic perspective, as the ITZ strength increases, the force-chain quantity and strength of the specimen will increase, and the contact anisotropy will become weaker. The specimen with low ITZ strength has a fragile and instable contact net, which resulted in crack development and contact anisotropy. Therefore, a relatively stable force-chain structure has a great influence on the mechanical properties of concrete.

In summary, the ITZ strength has a great impact on the mechanical properties of concrete when the surface is too small. Controlling the minimum quality of ITZ is the basic requirement to ensure the effectiveness of concrete.

## Figures and Tables

**Figure 1 materials-15-08840-f001:**
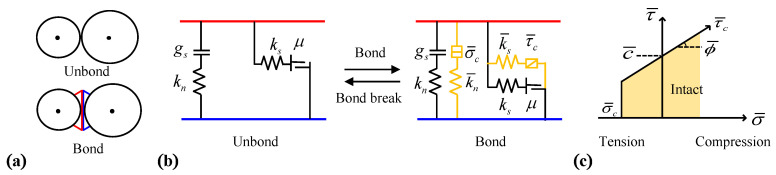
Parallel bond model. (**a**) Model diagram; (**b**) model theory; (**c**) strength criterion.

**Figure 2 materials-15-08840-f002:**
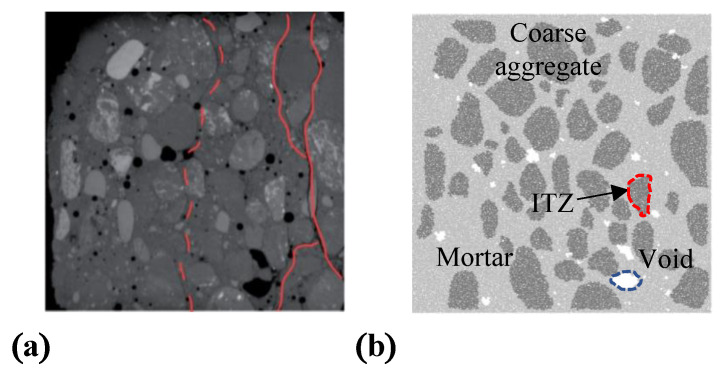
Concrete model. (**a**) Concrete specimen [15]; (**b**) concrete model.

**Figure 3 materials-15-08840-f003:**
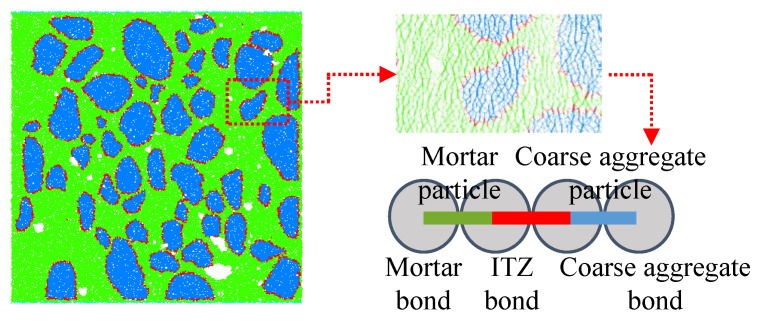
Bonds in concrete model.

**Figure 4 materials-15-08840-f004:**
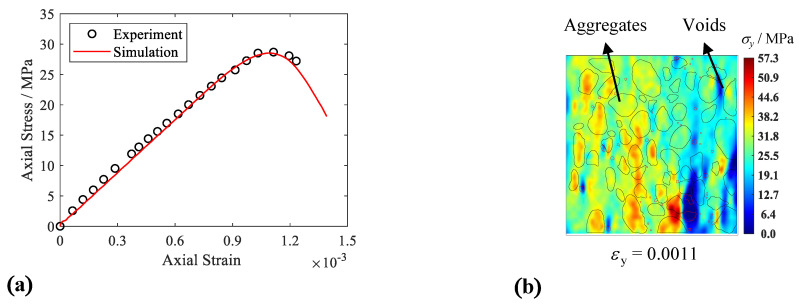
Concrete simulation results. (**a**) Stress—strain curve; (**b**) stress field at peak stress state; (**c**) crack number—strain curve; (**d**) failure surface.

**Figure 5 materials-15-08840-f005:**
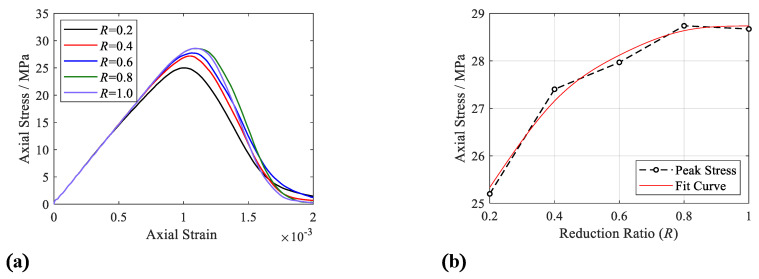
Mechanical properties. (**a**) Stress—strain curve; (**b**) peak stress summary graph.

**Figure 6 materials-15-08840-f006:**
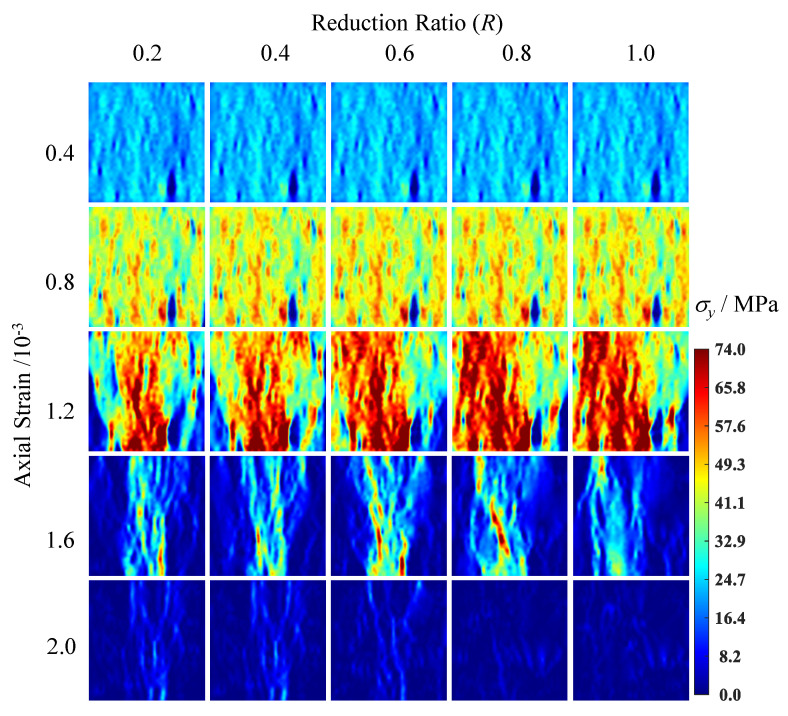
Stress field with different strain state.

**Figure 7 materials-15-08840-f007:**
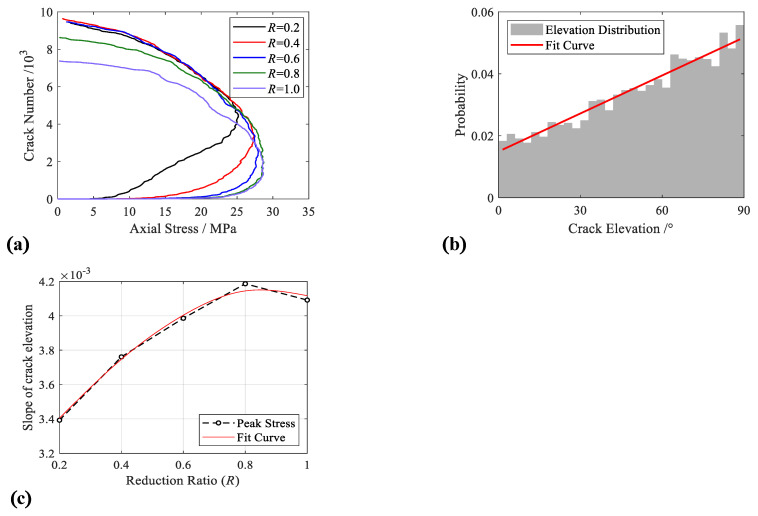
Crack properties. (**a**) Crack number—stress curve; (**b**) distribution of crack elevation (*R* = 1.0); (**c**) slope statistics of crack elevation.

**Figure 8 materials-15-08840-f008:**
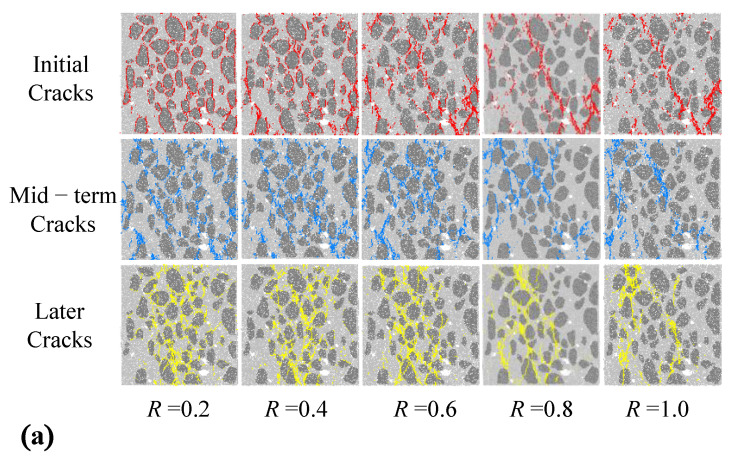
Spatial distribution of cracks. (**a**) Development of cracks in different stages; (**b**) distribution of shearing cracks and tensile cracks.

**Figure 9 materials-15-08840-f009:**
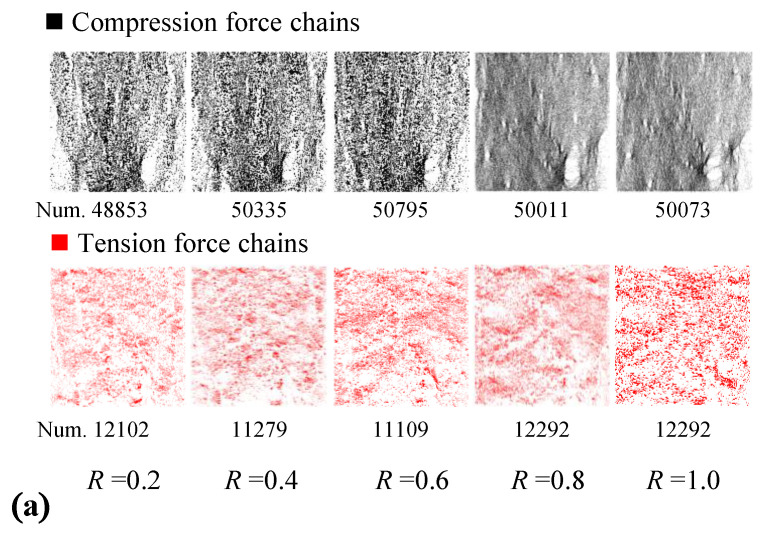
Force-chains strength properties. (**a**) Distribution of tension chains and compression chains; (**b**) strength probability of force—chains.

**Figure 10 materials-15-08840-f010:**
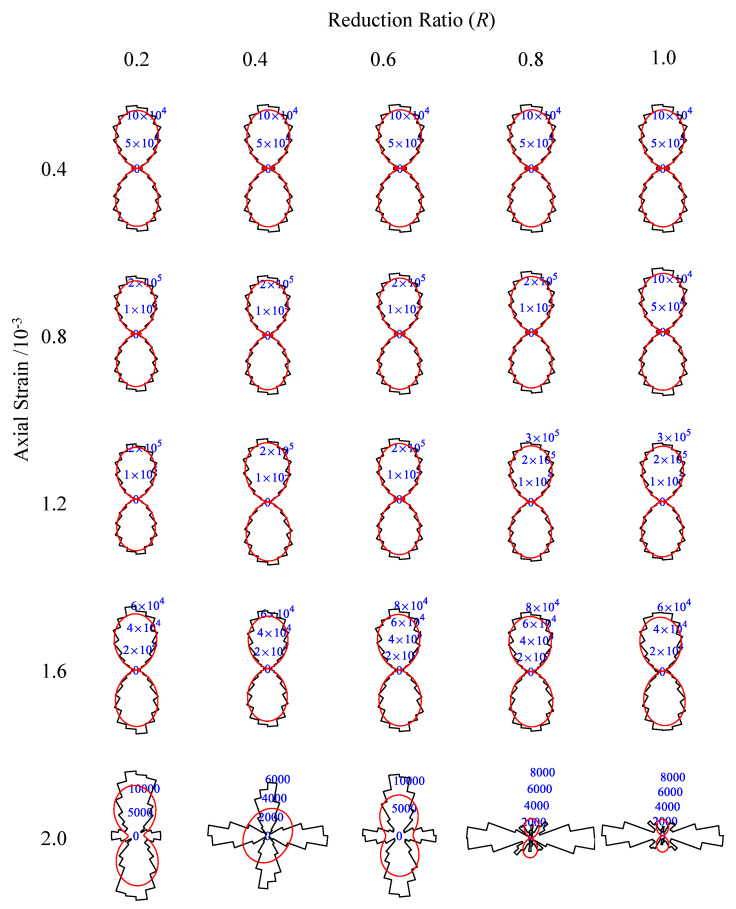
Summary of contact force—chains orientation.

**Figure 11 materials-15-08840-f011:**
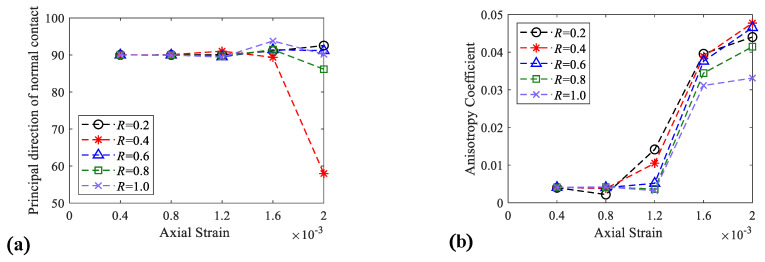
Contact force-chains orientation properties. (**a**) Force—chains angle axial—strain curve; (**b**) anisotropy coefficient axial—strain curve.

**Table 1 materials-15-08840-t001:** Parameters of concrete.

Materials	*E* (GPa)	*k*	$\bar{E}$ (GPa)	$\bar{k}$	${\bar{\sigma }}_{c}$ (MPa)	$\bar{c}$ (MPa)	$\bar{\phi }$ (°)
Coarse aggregate	33.40	1.50	33.40	1.50	29.50	29.50	40.00
Mortar	20.04	1.50	20.04	1.50	17.70	17.70	45.00
ITZ	16.03	1.50	16.03	1.50	14.16	14.16	35.00

**Table 2 materials-15-08840-t002:** Parameters scheme.

Materials	Test Number	*E* (GPa)	*k*	$\bar{E}$ (GPa)	$\bar{k}$	Reduction Factor/*R*	${\bar{\sigma }}_{c}$ (MPa)	$\bar{c}$ (MPa)	$\bar{\phi }$ (°)
Coarse aggregate	1, 2, 3, 4, 5	33.40	1.50	33.40	1.50	---	29.50	29.50	40.00
Mortar		20.04	1.50	20.04	1.50		17.70	17.70	45.00
ITZ	1	16.03	1.50	16.03	1.50	0.2	2.83	2.83	35.00
	2					0.4	5.67	5.67	
	3					0.6	8.50	8.50	
	4					0.8	11.33	11.33	
	5					1.0	14.16	14.16	

## Data Availability

All data, models, and code generated or used during the study appear in the published article.

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
