# Peer review of "Mesoscale Analysis of the Effect of Interfacial Transition Zone on the Compressive Damage of Concrete Based on Discrete Element Method"

_materials, 2022, doi:10.3390/ma15248840_

Round 1

Reviewer 1 Report

The authors proposed studying the influence of interfacial transition zone on mechanical properties of a concrete through numerical modeling – using discrete element method. In this case, concrete was divided into four phases: aggregates, mortars, ITZ and voids. In addition, the effects of ITZ were also investigated under different ITZ strength. The results show that with the increase in ITZ strength, the compressive strength of a concrete also increases. However, the number of cracks decreases, and the anisotropy of contact orientation becomes weaker. The manuscript is very well structured, well organized, and easy to follow. However, some information needs to be clarified in the manuscript. 

(1)   Description of information from several papers on the ITZ behavior and numerical modeling is very interesting. However, the justification needs to be improved. What is the novelty of the study? What are you doing differently from other studies? 

(2)   Lines 74-75: “The parallel bond model (PBM) developed by D.O.Potionand P.A.Cundall [20] included bonded and unbonded parts, (Fig.1(a))” were used in this present study. Please, add a complement at the end of the sentence. 

(3)   Section 2.2: Was the compression strength test performed experimentally or during the numerical simulation? 

(4)   Line 96: The authors specified four phases in concrete: mortar, aggregate, ITZ and void. However, a mortar also has aggregate. Make it clear that the focus is on the coarse aggregate which is responsible for the ITZ.  

(5)   Lines 108 and 111: Please, correct Figure numbers. 

(6)   Line 108: Figure 1 and 2: Please, replace “aggregate” with “coarse aggregate”. 

(7)   Please type the units between () instead of / 

(8)   What is the explanation for the maximum stress in stress nephograms being twice the maximum stress in the stress-strain curves? 

(9)   Figure 9 (a): It is difficult to see color red in the figure.

 (10)                   There is no discussion section. It is suggested that the authors link the results observed for different properties considered in this study with more bibliographic references. The discussion section must be further improved.

Author Response

Point 1: Description of information from several papers on the ITZ behavior and numerical modeling is very interesting. However, the justification needs to be improved. What is the novelty of the study? What are you doing differently from other studies?

Response 1: At present, the main analysis points of ITZ effect are focus on macro mechanical properties[1-3], such as stress-strain curve, destruction, etc., and there is a lack of analysis on the internal contact force of granular materials, the novelty of the study is to quantitatively analyze the influence of ITZ strength on the contact force chain network from micro perspective. Therefore, the model we built considers the real concrete failure surface which can reflect a real contact force chain network, and we analyzed the ITZ effect step by step from macro, micro and micro perspectives in section 4.

We modified the last paragraph of introduction.

Point 3: Section 2.2: Was the compression strength test performed experimentally or during the numerical simulation?

Response 3: We modified this sentence.

Herein, the axial laboratory compression test on the concrete specimen with a size of 50´50´50 mm3 was selected as a reference experiment.

Point 4: Line 96: The authors specified four phases in concrete: mortar, aggregate, ITZ and void. However, a mortar also has aggregate. Make it clear that the focus is on the coarse aggregate which is responsible for the ITZ.

Response 4: We have corrected all inaccuracies in the paper. See the revised manuscript for details.

Point 5: Lines 108 and 111: Please, correct Figure numbers.

Response 5: We have proofread.

Point 7: Please type the units between () instead of /

Response 7: We corrected them accordingly.

Point 8: What is the explanation for the maximum stress in stress nephograms being twice the maximum stress in the stress-strain curves?

Response 8: The stress in the stress-strain curve is an average value at both ends of the specimen, the stress nephogram shows the stress inside the specimen, both are at a same strain, but the internal maximum stress can reach twice the average stress. We revised the expression of original paper for clarity.

Fig. 4(b) is the nephogram of the axial stress (sy) field in the specimen at an axial strain level of 0.0011, at which the stress in stress-strain curve reaches maximum of 28.7 MPa. The stress distribution in specimen is inhomogeneous, and the maximum stress zone accumulated around the outlines of the coarse aggregate, shown in red, is 57.3 MPa. Compared with the peak stress in stress-strain curve, the maximum stress in the specimen can reach twice peak stress.

Point 9: Figure 9 (a): It is difficult to see color red in the figure.

Response 9: Because the strength and quantity of tension force chains in compression test are very small. To be clearly, we redraw the figure.

Point 10: There is no discussion section. It is suggested that the authors link the results observed for different properties considered in this study with more bibliographic references. The discussion section must be further improved.、

Response 10:We revised the conclusion as following:

In this paper, a four-phase concrete model based on the real damage surface of concrete was established by PFC2D. The model can reflect the mechanical properties of concrete well by considering the effects of ITZ strength. On this basis, compression tests were carried with different ITZ strength coefficients.

In the tests, macro mechanical strength curves are first studied, the peak stress of concrete increases as the ITZ strength increases, and the corresponding effects can be ignored after R≥0.8, which is same as previous studies[2, 4], Further study of stress nephogram shows that the stress in specimen is inhomogeneous and accumulate around outlines of coarse aggregate, and maximum stress in specimen is around two times the peak stress of specimen.

The micro crack and contact characteristics are subsequently studied, with the increase in ITZ strength, the initial cracks are transformed from an ITZ zone into a void zone, the crack’s number, and elevation increase; meanwhile, the concrete is prone to brittle failure and the concrete will be damaged from shearing to tensile. For microscopic perspective, as the ITZ strength increases, the force-chain quantity and strength of specimen will increase, and the contact anisotropy will become weaker. Specimen with low ITZ strength has a fragile and instable contact net, which resulted in crack development and contact anisotropy. Therefore, a relatively stable force chain structure has a great influence on the mechanical properties of concrete.

In summary, the ITZ strength has a great impact on the mechanical properties of concrete when is too small. Controlling the minimum quality of ITZ is the basic requirement to ensure the effectiveness of concrete.

Reviewer 2 Report

The manuscript presents a simulation method for analysis of concrete failure due to compression. The manuscript is clear, short and up-to-the-point, making it very easy to follow and to read, as it is also present a good structure. The introduction -without being loquacious- states directly the problem and is well supported by literature. The analysis concludes to the results of the model. I have just one think to point out, which I believe that it will strengthen the scientific soundness of this study:

1.       Please, enrich your conclusion section by showing the relativeness of applying DEM in the “concrete” case. Ok, presenting the main results is the obvious think to do, but something should be mentioned regarding the applied simulation method. For example, are the results comparable with previous studies?              

Apart from this, the manuscript is well written in terms of language. Some minor points to take care:

·         Line 21. Decreases

·         Line 54. Please change to: “discrete element method (DEM)”

·         Line 180: “additional statistical analysis … was”

·         Line 212: “where the distribution … is clear”

Author Response

Response 1:We revised the conclusion as following:

In this paper, a four-phase concrete model based on the real damage surface of concrete was established by PFC2D. The model can reflect the mechanical properties of concrete well by considering the effects of ITZ strength. On this basis, compression tests were carried with different ITZ strength coefficients.

In the tests, macro mechanical strength curves are first studied, the peak stress of concrete increases as the ITZ strength increases, and the corresponding effects can be ignored after R≥0.8, which is same as previous studies[17, 29], Further study of stress nephogram shows that the stress in specimen is inhomogeneous and accumulate around outlines of coarse aggregate, and maximum stress in specimen is around two times the peak stress of specimen.

The micro crack and contact characteristics are subsequently studied, with the increase in ITZ strength, the initial cracks are transformed from an ITZ zone into a void zone, the crack’s number, and elevation increase; meanwhile, the concrete is prone to brittle failure and the concrete will be damaged from shearing to tensile. For microscopic perspective, as the ITZ strength increases, the force-chain quantity and strength of specimen will increase, and the contact anisotropy will become weaker. Specimen with low ITZ strength has a fragile and instable contact net, which resulted in crack development and contact anisotropy. Therefore, a relatively stable force chain structure has a great influence on the mechanical properties of concrete.

In summary, the ITZ strength has a great impact on the mechanical properties of concrete when is too small. Controlling the minimum quality of ITZ is the basic requirement to ensure the effectiveness of concrete.

Point 2: Apart from this, the manuscript is well written in terms of language. Some minor points to take care:

  • Line 21. Decreases
  • Line 54. Please change to: “discrete element method (DEM)”
  • Line 180: “additional statistical analysis … was”
  • Line 212: “where the distribution … is clear”

Response 2We corrected them accordingly.

Round 2

Reviewer 1 Report

Thank you for addressing all my comments